# Effects of Interleukin-1 Genotype on the Clinical Efficacy of Non-Surgical Periodontal Treatment of Polish Patients with Periodontitis

**DOI:** 10.3390/biomedicines11020456

**Published:** 2023-02-04

**Authors:** Aniela Brodzikowska, Bartłomiej Górski, Agnieszka Bogusławska-Kapała

**Affiliations:** 1Department of Conservative Dentistry, Medical University of Warsaw, ul. Binieckiego 6, 02-097 Warszawa, Poland; 2Department of Periodontology and Oral Mucosa Diseases, Medical University of Warsaw, ul. Binieckiego 6, 02-097 Warszawa, Poland; 3Department of Comprehensive Dental Care, Medical University of Warsaw, ul. Binieckiego 6, 02-097 Warszawa, Poland

**Keywords:** periodontitis, IL-1 genotype

## Abstract

(1) Background: Periodontitis is a chronic multifactorial inflammatory disease associated with dysbiotic plaque biofilms and characterized by progressive destruction of the tooth-supporting apparatus. The aim of the study was to evaluate the efficacy of basic periodontal treatment depending on the interleukin-1 genotype in adult Poles. (2) Methods: Sixty subjects aged 39–64 years were examined. At initial presentation (T1), at 6–8 weeks (T2), and 16–18 weeks (T3) after treatment completion, the following percentages were recorded: surfaces with plaque, pockets bleeding, pocket depth, and change in the attachment level. During the T1 examination, the genotype for IL-1 was determined using the GenoType^®^ PST test. (3) Results: Thirty subjects had genotype IL+ and the other thirty were IL−. During the T1 examination no significant differences were observed between patients. The study showed an increase of all the tested clinical parameters after 6–8 weeks. This increase continued up to the T3 examination. A significant reduction in the percentage of plaque surfaces after 6–8 weeks was observed, which was sustained after 16–18 weeks for both genotypes. For both genotypes, a significant decrease in the percentage of bleeding pockets was observed at the T2 examination, which persisted through until examination T3. For both studied genotypes, after 6–8 weeks, a significant shallowing of pockets was observed. In patients with the IL− genotype, a further significant shallowing of pockets was observed after 16–18 weeks. A significant reconstruction of epithelial attachment was observed between the T1 and T2 examinations, averaging 0.55 mm in patients with the IL+ genotype, and 0.77 in patients with the IL− genotype. (4) Conclusions: The results of our study show that the IL-1 genotype, may be one of the factors affecting the healing process after non-surgical periodontal treatment in adult Poles.

## 1. Introduction

Clinical symptoms of periodontitis result from a chronic inflammatory process in host’s tissues, in response to presence of pathogenic microorganisms. The mechanisms of inflammation involve numerous proinflammatory mediators; hence, an increased risk of periodontitis may be related to an aberrant production of proinflammatory cytokines [1]. Interleukin-1 (IL-1) appears to be one of the most important molecules that is associated with the development and persistence of periodontal inflammation [2]. Interleukin-1 (IL-1) was first cloned in the 1980s and rapidly emerged as a key player in the regulation of inflammatory processes. The term IL-1 refers to two cytokines, IL-1 alpha and IL-1 beta, which are encoded by two separate genes. Genetic variations in interleukins (IL) genes are associated with chronic periodontitis [3]. Interferon gamma (IFN-γ) is also an immunoregulatory cytokine that plays an important role in the activation of inflammatory processes, which are the basis of periodontal disease. Interferon-γ regulates stem cells in the periodontal ligament which have immunomodulatory capacities [4].

Clinical studies have shown considerable individual differences in susceptibility to periodontitis, in its clinical picture and in the course of the disease [5]. In addition, significant differences in the effectiveness of the disease treatments between patients were observed [6]. Currently it is believed that a vital role in the formation and development of periodontitis is played by factors influencing the defensive abilities of the host, such as genetic predisposition, some systemic diseases (especially diabetes), and environmental factors, particularly smoking [7]. Evidence provided by periodontal research on genetic risk factors is of uttermost importance in clinical practice as a possible diagnostic and prognostic tool for periodontitis.

Periodontal disease has systemic implications in overall health homeostasis. Periodontitis involves dysbiotic events caused by pathogenic microflora that systemically exacerbate the proinflammatory status. Diabetes mellitus, cardiovascular disease, hypertension, metabolic syndrome and obesity, hypertension, chronic kidney disease, chronic obstructive pulmonary disease, and autoimmune diseases have been identified as risk factors for periodontitis and severe COVID-19 infections. These diseases have similar inflammatory pathways that are involved in the progression of the conditions [8].

In addition, rheumatoid arthritis is a chronic inflammatory disease sharing genetic and environmental risk factors similar to periodontitis. These two diseases have polymorphisms in genes that encode certain cytokines which results in connective tissue damage and bone alterations. Periodontitis and rheumatoid arthritis are manifested as persistent levels of proinflammatory cytokines and associated molecules. Therapeutic strategies in rheumatoid arthritis based on blocking proinflammatory cytokines have had an impact on the overall periodontal status [9].

One of the directions of research aimed at identifying individuals with an increased risk of periodontitis is an attempt to determine polymorphisms of genes controlling or modifying different aspects of the host response. Genetic polymorphisms associated with increased production of pro-inflammatory mediators may potentially contribute to greater severity of periodontitis. In 1997, Kornman et al. [5] described a gene polymorphism accompanied by increased production of interleukin-1 (IL-1) which was used as a “marker” of a tendency to develop severe forms of chronic periodontitis in adults.

The polymorphism is characterized by the presence of allele 2 in both the IL-1A gene, encoding IL-1α, at locus −889, as well as in the IL-1B gene, encoding IL-1β, at locus +3953 [10]. It was found that in individuals with the mentioned genotype, defined by Kornman et al. [5] as a positive genotype, chronic periodontitis was more common, which was also characterized by a more severe course.

The aim of the study is to assess the effectiveness of non-surgical periodontal therapy in adult Polish patients with periodontitis, depending on IL-1 genotype. The null hypothesis was that a patient with a particular phenotype responds differently to treatment.

## 2. Materials and Methods

The present study design and protocol were approved by the institutional review board (KB/58/2011). All clinical procedures were performed in accordance with the Helsinki Declaration of 1975, as revised in Tokyo in 2013. The participating subjects were recruited among periodontitis patients referred to the Department of Conservative Dentistry of the Medical University of Warsaw for periodontal treatment. (Figure 1). After having been informed about the study’s objectives, as well as the possible risks and benefits of participating in the study, all patients gave informed consent.

### 2.1. Inclusion Criteria

The inclusion criteria were as follow: (1) systemically healthy Polish volunteers, diagnosed with stage I, II, III, IV grade B periodontitis; (2) presence of ≥20 teeth; (3) no professional teeth cleaning within 6 months before the examination; and (4) agreed to participate in the study and signed a written consent form. 

### 2.2. Exclusion Criteria

The exclusion criteria were as follows: (1) systematic diseases that affect the appearance and progress of periodontal disease (e.g., diabetes, blood disorders, or immunodeficiency); (2) medication affecting healing of periodontal and peri-implant tissues (e.g., antibiotics, steroids, anti-inflammatory drugs, immunosuppressants, antiepileptic drugs, and calcium channel blockers); (3) pregnancy or lactating; and (4) smokers or user of other tobacco products.

### 2.3. Clinical Examination

The study consisted of a clinical and laboratory part. A periodontal examination was performed by a single and previously calibrated examiner (A.B.). A total of 10 non-study patients were recruited for the calibration exercise. The designated examiner recorded full-mouth PPD and CAL with an interval of 24 h between recordings. Calibration was accepted when ≥90% of the recordings were reproduced within a difference of 1.0 mm and an exact agreement was repeated in 75% of measurements. The following clinical parameters were evaluated with a calibrated periodontal probe (PCPUNC156, Hu-Friedy^®^, Chicago, IL, USA) with a diameter of 0.5 mm: (1) the number of teeth present in the oral cavity; (2) full-mouth plaque index (FMPI) according to O’Leary et al. [11] as presence or absence (4 sites per tooth); (3) full mouth bleeding on probing index (FMBOP) according to Ainamo and Bay [12] and using the same probing pressure, sulcus bleeding was determined 30 s after probing and was assessed as either presence or absence at six sites per tooth (i.e., distobuccal, buccal, mesiobuccal, distolingual, lingual, mesiolingual); (4) probing pocket depth (PPD) by applying a calibrated periodontal probe at six points on each tooth as the distance from the gingival margin to the bottom of the pocket; and (5) clinical attachment level (CAL) that was recorded at six points on each tooth as the distance from the cemento-enamel junction (CEJ) to the bottom of the pocket. Radiographic changes were measured from standardized periapical radiographs and panoramic X-rays. Stage I periodontitis was diagnosed when: (1) interdental CAL 1–2 mm; (2) radiographic bone loss of 15%; (3) without tooth loss; and (4) PPD < 4 mm. Stage II periodontitis was diagnosed when: (1) interdental CAL 3–4 mm; (2) radiographic bone loss of 15%, to the mid-third of the tooth root; (3) without tooth loss; and (4) PPD < 5 mm. Stage III periodontitis was diagnosed when: (1) interdental CAL ≥ 5 mm; (2) radiographic bone loss extended to the mid-third of the tooth root and beyond; (3) tooth loss due to periodontitis was ≤4 teeth; and (4) PPD ≥ 6 mm. Stage IV periodontitis was recognized when: (1) interdental CAL ≥ 5 mm; (2) radiographic bone loss extended to the mid-third of the tooth root and beyond; (3) tooth loss due to periodontitis was ≥5 teeth; and (4) PPD ≥ 6 mm. Grade B periodontitis was rated indirectly on dental radiograms as percentage of root length divided by the age of the subject (% bone loss/age) when this value varied from 0.25 to 1.0. Healthy periodontium was evaluated as <10% bleeding sites with PPD ≤ 3 mm [13]. 

Each patient was examined three times: 

The first examination (T1) was a preliminary clinical examination. The following clinical parameters were evaluated: the number of teeth, FMPI, FMBOP, PPD, CAL panoramic and periapical X-rays, and stage of periodontitis. 

The second (T2) was conducted after 6–8 weeks. The following clinical parameters were evaluated: FMPI, FMBOP, PPD, and CAL. 

The third examination (T3) was performed after 16–18 weeks. The following clinical parameters were evaluated: FMPI, FMBOP, PPD, and CAL.

### 2.4. Genetic Examination

During the T1 examination a buccal swab was taken to determine the genotype for IL-1 using GenoType^®^ PST (Hain Lifescience GmbH, Nehren, Germany). The test consists of isolating DNA from the obtained epithelial cells and then amplification of IL-1A and IL-1B gene fragments with the polymerase chain reaction method. The obtained DNA fragments underwent reverse hybridization with probes identifying alleles at loci IL-1A^−889^ and IL-1B^+3953^. On the basis of this examination, patients were divided into two groups: the positive genotype (IL+) group and negative genotype (IL-) group.

### 2.5. Periodontal Treatment

All participants received non-surgical periodontal treatment that was carried out by the same periodontist. It consisted of full-mouth scaling and root planning (SRP) performed in a single appointment under local anesthesia using an ultrasonic device and hand instruments. Oral hygiene instructions included Bass brushing technique and use of an interdental brush and dental floss.

### 2.6. Statistical Analysis

Statistical analysis was performed using Statistica v. 13 (TIBCO Software Inc., Palo Alto, CA, USA). Mean and standard deviations were calculated for each parameter. The normal distribution was analyzed with the Shapiro–Wilk test. The numeric demographic parameters that showed a normal distribution were analyzed by a *t*-test. The mean values of the assessed parameters before and after treatment were compared using the Wilcoxon signed-rank test and between patients with the IL+ genotype and patients with the IL− genotype the Mann–Whitney U test was used. The level of significance was *p* = 0.05.

To assess the correlation between the clinical indices of periodontitis and the interleukin-1 polymorphism, the Spearman’s correlation coefficient was used.

## 3. Results

The study included 60 patients with stage I, II, III, and IV periodontitis grade B (42 women, 18 men) aged 39–64 years. The mean age of the patients was 55.2 ± 4.2. A total of 30 patients had the positive genotype IL+ (20 women, 10 men) and 30 had the negative genotype IL− (22 women, 8 men). All measurements were performed by an experienced and calibrated clinician who was blinded with respect to the clinical intervention. The test subjects were calibrated at each stage. The demographic and clinical characteristics are presented in Table 1.

During the T1 examination no significant differences were observed between patients with different genotypes for any of the tested parameters, or in connection with age differences between the patients. The clinical parameters (mean and standard deviation) at baseline, 6–8 weeks, and 16–18 weeks after treatment are shown in Table 2. The study showed improvement of all the tested clinical parameters after 6–8 weeks. This improvement continued up to the T3 examination after 16–18 weeks. When comparing the percentages of surfaces with dental plaque (Table 2), the authors observed a significant reduction in the percentage of plaque surface after 6–8 weeks (T2) for both genotypes, which was sustained after 16–18 weeks (T3). The small decrease in the percentage of plaque surfaces between examinations T2 and T3 was not statistically significant. No significant difference in the percentage of surfaces with plaque between patients with IL+ genotype and patients with IL− genotype was observed at any point during the study.

For both studied genotypes, a significant reduction in the percentage of bleeding pockets was observed at the T2 examination, which persisted through until examination T3 (Table 2). There was no significant change between examination T2 and T3. A comparison between the patients with the IL+ genotype and the ones with the IL− genotype showed no significant differences at any point during the study.

For both studied genotypes, after 6–8 weeks, a significant shallowing of pockets was observed (Table 2). In patients with the IL− genotype, a further significant shallowing of pockets was observed after 16–18 weeks, while in patients with the IL+ genotype, further shallowing was not significant during the T3 examination. For both genotypes, a significant increase in the percentage of pockets < 4 mm deep was observed between the T1 and T2 examinations (Table 2). In addition, between the T2 and T3 examinations, a further increase in the percentage of shallow pockets was observed for both genotypes, which reached a statistically significant level in patients with the IL− genotype. Comparing the percentage of shallow pockets between patients with IL+ and IL− genotypes showed comparable values of this parameter in both groups during Examinations T1 and T2, while in Examination T3 there was a much higher percentage of pockets < 4 mm deep in patients with genotype IL−. For both genotypes, a decreased percentage of pockets 4–6 mm deep between the T1 and T2 examinations was observed (Table 2). In addition, between the T2 and T3 examinations for both genotypes there was a further slight decrease in the percentage of medium-deep pockets; however, the difference was not significant. A comparison of the percentage of medium-deep pockets between patients with IL+ and IL− genotypes showed similar values of this parameter in both groups during Examinations T1 and T2, while in the T3 examination a much larger percentage of pockets 4–6 mm deep in patients with the IL+ genotype was observed.

For both genotypes, a significant reduction in the percentage of pockets > 6 mm deep was observed between the T1 and T2 examinations (Table 2). In addition, between the T2 and T3 examinations, a further slight decrease in the percentage of deep pockets was observed, but the difference between the T2 and T3 examinations was significant only in patients with the IL+ genotype. At no point during the study was there was a significant difference in the percentage of deep pockets between patients with IL+ and patients with IL−.

For both genotypes, a significant reconstruction of epithelial attachment was observed between the T1 and T2 examinations, averaging 0.55 mm in patients with the IL+ genotype and 0.77 mm in patients with the IL− genotype (Table 2). A further reconstruction of attachment between Examinations T2 and T3 was 0.24 mm in patients with the IL+ genotype (*p* > 0.05) and 0.17 mm in patients with the IL− genotype (*p* < 0.05).

In the group of patients diagnosed with periodontitis and with the IL+ genotype, we observed a strong positive correlation between the IL+ genotype and the probing depth reduction (r = 0.373 ÷ 0.441; *p* < 0.05) overall. On the other hand, there was no correlation in the studied groups between the IL-1 SNP (single nucleotide polymorphism) and BOP and the plaque index (r = 0.13; *p* > 0.05; r = 0.162; *p* > 0.05, respectively). The highest correlation coefficients were found between the IL+ SNP and the deep periodontal pocket reduction in patients with severe periodontitis (stage III and IV, r = 0.638; *p* < 0.05 and r = 0.549; *p* < 0.05, respectively).

## 4. Discussion

The treatment results were comparable with the results obtained by other authors who evaluated the effectiveness of non-surgical treatment in studies shorter than 6 months. The study showed a reduction in the percentage of dental surfaces with dental plaque from 85% to 53% after 6–8 weeks, which was maintained at this level up to 16–18 weeks of observation. The results in this study were within the range of values observed by authors of earlier works, in which dental plaque reduction was found to be from 34–100% at baseline to 10–65% after three months [14,15,16].

The lack of significant improvement between 6–8 weeks and 16–18 weeks is consistent with the results obtained by Yan et al. [17] and Smiley et al. [18], who, one week after scaling and hygiene instruction, found a significant reduction in the amount of plaque, which remained at the same level for the next 3 and 15 weeks of observation. Albonni et al. [19] found the greatest improvement in oral hygiene during the first two weeks after treatment, with little changes after 8 and 16 weeks, despite repeated instructions on oral hygiene and professional tooth cleaning.

Our research showed that the IL-1 genotype affected neither the percentage of tooth surfaces covered with dental plaque, nor improvements in oral hygiene after treatment. Similarly, Meisel et al. [20] in their epidemiological study found no correlation between the IL-1 genotype and oral hygiene status. De Sanctis and Zucchelli [21], on the other hand, found no difference in the percentage of surfaces with plaque between patients with IL+ and IL− genotypes after the initial treatment of periodontitis and also one and four years after surgical regenerative treatment.

After 6–8 weeks, in parallel with a decreased percentage of surfaces with plaque, a reduction of pockets bleeding on probing from 52% to 19% was observed. This result was unchanged until the end of the research. The obtained results were within the ranges reported by other authors who observed a reduction in the percentage of bleeding pockets from 33–93% (at baseline) to 4–61% (after 6–9 weeks) [2,22]. Povsic et al. [23] also did not observe any further significant change during 2 to 4 months of follow-up, nor did Walther et al. [24] after 1 to 6 months and Pozo et al. [25] between 2 and 6 months after treatment.

No impact of the IL-1 genotype on the percentage of pockets bleeding on probing was observed, as well as on its reduction after treatment. Similarly, Meisel et al. [20] did not find correlations between IL-1 genotype and pocket bleeding in their epidemiological study, nor did Kornman et al. [5] and Liu et al. [1] in patients with untreated periodontitis, Jepsen et al. [26] in individuals neglecting hygiene procedures, or De Sanctis and Zucchelli [21] in patients under maintenance treatment after surgical regenerative treatment. On the other hand, Lang et al. [27] found a relationship between the presence of the IL+ genotype and pocket bleeding in patients undergoing maintenance therapy, both in a cross-sectional study, as well as in a follow-up after approximately one-year. This correlation was observed only in individuals who had never smoked tobacco.

Between 6 to 8 weeks after treatment, some shallowing of the periodontal pockets was observed, which was expressed by both reduced average depth of pockets, as well as an increased percentage of shallow pockets and decreased percentage of medium-deep and deep pockets. The improvement was maintained during 16–18 weeks of observation. The improvement obtained in the present study between recall visits after 6–8 weeks and examinations after 16–18 weeks confirmed the results obtained by Walther et al. [24], who observed a shallowing of pockets by 0.84 mm after one month and by a further 0.15 mm after another 2 months. However, Albonni et al. [19] found no change between recalls after 8 weeks and 16 weeks, similar to Nile et al. [28] between recalls after 6 weeks and after 3 months.

In this study no influence of the IL-1 genotype on the mean depth of periodontal pockets was noted. 

On the other hand, in patients with the IL− genotype, a significant further shallowing of pockets between the examinations after 8 weeks and 16 weeks was observed, which was accompanied by an increase in the percentage of shallow pockets (<4 mm) to a level exceeding that observed in patients with the IL+ genotype, with a simultaneous decrease in the percentage of medium-deep pockets (4–6 mm). In the case of deep pockets (>6 mm), high standard deviations in the percentages of pockets in this range were observed; additionally, a comparison between patients with IL+ and IL genotypes demonstrated no significant differences between these groups. Similar to this study, Liu et al. [1] also found no correlation between IL-1 genotype and mean pocket depth as well as the percentage of deep pockets in patients with untreated periodontitis. Likewise, Meisel et al. [20] did not find such a relationship in a randomly selected group of individuals. Kornman et al. [4,5] and Cullinan et al. [29] found greater mean pocket depths in patients with the IL+ genotype than in patients with the IL− genotype among non-smokers, but among smokers, patients with the IL+ genotype demonstrated a higher percentage of pathological pockets.

Lang et al. [27] found no correlation between IL-1 genotype and the number of persistent pathological pockets among patients under maintenance treatment. In addition, Caffesse et al. [30,31] observed no relationship between genotype and mean pocket depth, both in a combined analysis of smokers and non-smokers as well as when assessing non-smokers only. Meanwhile, De Sanctis and Zucchelli [21] observed a group of patients under maintenance treatment after guided tissue regeneration. The authors found that despite a comparable improvement in patients with IL+ and IL– genotypes one year after the procedure, at the four-year recall, a greater deepening of pockets was observed in IL+ genotype patients.

Parallel to the shallowing of the pockets, after 6–8 weeks, restoration of attachment was observed, which lasted for 16–18 weeks of observation. As with pocket depth measurements, significant further reconstruction of attachment was observed between examinations after 6–8 weeks and examinations after 16–18 weeks in patients with the IL− genotype. The improvement obtained in the present study between examinations after 6–8 weeks and examinations after 16–18 weeks is consistent with observations of Smiley et al. [18], who reported reconstruction of attachment by 0.63 mm after 8 weeks and then by another 0.07 mm after 16 weeks. After surgical regenerative treatment, Christgau et al. [32] as well as De Sanctis and Zucchelli [21] observed similar attachment reconstruction in the one-year follow-up for both genotypes. The studied patients were included in a program of regular control visits every month or every 2–3 months. For the next 3 years, during which the frequency of follow-up visits was reduced, De Sanctis and Zucchelli [21] observed a loss of the obtained attachment in comparison with the one-year recall visits. The abovementioned loss was greater in patients with the IL+ genotype. The authors recommend a more rigorous maintenance treatment program for patients with the IL+ genotype in order to sustain the effects of surgery.

Both in relation to pocket depths and attachment level, the authors evaluating the efficacy of basic treatment observed the greatest clinical improvement 1–3 months after scaling, then a plateau was reached with only slight changes in the examined parameters after 6 months. However, it was assumed that the periodontal healing and regeneration process could continue during the following 9 to 12 months [16].

Although a comprehensive investigation of the effects of the interleukin-1 genotype on the clinical efficacy of basic periodontal treatment has been performed, there were some limitations to this work. First, there was a limited number of included subjects. The group only contained systematically healthy and non-smoking subjects. Second, due to the fact that there were pilot studies, there was no sample size calculation. It is well known that the sample size may influence the genetic factors. The results of this study will be used to calculate the sample in future larger investigations.

## 5. Conclusions

The purpose of this work was to shed light on the influence of genetic conditions on the results of the basic periodontal treatment. The results of our research indicate that the IL-1 genotype may be one of the factors affecting the healing process after non-surgical periodontal treatment between 6–8 and 16–18 weeks subsequent to the procedure, contributing to further shallowing of pockets and reconstruction of attachment during this period in adult Polish patients with the IL− genotype and to a lack of improvement in patients with the IL+ genotype. This work is an important starting point for more research studies to better understand the role of the interleukin-1 genotype in the treatment of periodontal disease.

## Figures and Tables

**Figure 1 biomedicines-11-00456-f001:**
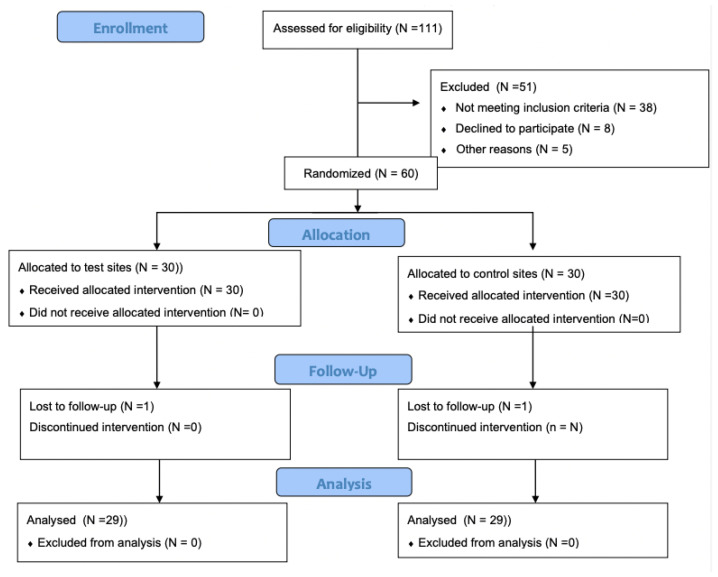
Consort diagram showing the study outline.

**Table 1 biomedicines-11-00456-t001:** Characteristics of the studied groups.

Variable	IL+	IL−	*p*
Age, mean ± SD (years)	4.3 ± 5.3	56.1 ± 4.2	0.432
Sex, F/M n (%)	20/10 (66.6/33/3)	22/8 (73.3/26.6)	0.453
Number of teeth, mean ± SD	21.56 ± 1.62	22.53 ± 1.30	0.456
FMPI, mean ± SD (%)	83.71 ± 10.76	86.91 ± 10.91	<0.001
FMBOP, mean ± SD (%)	52.78 ± 21.6	50.95 ± 28.09	<0.001
PPD, mean ± SD (mm)	3.55 ± 0.73	3.47 ± 1.12	<0.001
CAL, mean ± SD (mm)	9.85 ± 1.81	10.66 ± 2.13	<0.001
Stage I, n (%)	0 (0)	0 (0)	>0.05
Stage II, n (%)	10 (33.3)	9 (30)	>0.05
Stage III, n (%)	12 (40)	14 (46.6)	>0.05
Stage IV, n (%)	8 (26.6)	7 (23.3)	>0.05

SD—standard deviation; FMPI—full mouth plaque index; FMBOP—full mouth bleeding on probing index; PPD—probing pocket depth; CAL—clinical attachment loss.

**Table 2 biomedicines-11-00456-t002:** Clinical parameters (mean and standard deviation) at baseline (T1), 6–8 weeks (T2), and 16–18 weeks (T3) after treatment.

Variable	T1	T2	T3	P T1-T2	P T1-T3	P T2-T3
FMPI IL+ (%)	83.71 ± 10.76	56.61 ± 17.33	56.16 ± 17.17	*p* < 0.05	*p* < 0.05	*p* > 0.05
FMPI IL− (%)	86.91 ± 10.91	49.11 ± 21.36	45.77 ± 18.79	*p* < 0.05	*p* < 0.05	*p* > 0.05
*p*	*p* > 0.05	*p* > 0.05	*p* > 0.05			
FMBoP IL+ (%)	52.78 ± 21.6	21.79 ± 13.89	22.53 ± 13.06	*p* < 0.05	*p* < 0.05	*p* > 0.05
FMBoP IL− (%)	50.95 ± 26.09	17.1 ± 15.0	16.69 ± 13.51	*p* < 0.05	*p* < 0.05	*p* > 0.05
*p*	*p* > 0.05	*p* > 0.05	*p* > 0.05			
PPD IL+ (mm)	3.55 ± 0.73	2.7 ± 0.69	2.6 ± 0.56	*p* < 0.05	*p* < 0.05	*p* > 0.05
PPD IL− (mm)	3.47 ± 1.12	2.52 ± 1.01	2.33 ± 0.91	*p* < 0.05	*p* < 0.05	*p* < 0.05
*p*	*p* > 0.05	*p* > 0.05	*p* > 0.05			
CAL IL+ (mm)	9.85 ± 1.81	9.3 ± 2.01	9.06 ± 1.86	*p* < 0.05	*p* < 0.05	*p* > 0.05
CAL IL− (mm)	10.66 ± 2.13	9.89 ± 1.87	9.72 ± 1.88	*p* < 0.05	*p* < 0.05	*p* < 0.05
*p*	*p* < 0.05	*p* < 0.05	*p* < 0.05			
%PPD < 4 mm IL+	53.92 ± 19.03	77.41 ± 18.07	80.18 ± 14.93	*p* < 0.05	*p* < 0.05	*p* > 0.05
%PPD < 4 mm IL−	58.84 ± 23.95	82.85 ± 20.93	85.74 ± 19.54	*p* < 0.05	*p* < 0.05	*p* < 0.05
*p*	*p* < 0.05	*p* < 0.05	*p* < 0.05			
%PPD 4–6 mm IL+	40.21+/14.89	20.26 ± 15.93	18.45 ± 13.84	*p* < 0.05	*p* < 0.05	*p* > 0.05
%PPD 4–6 mm IL−	33.7 ± 16.41	13.9 ± 13.88	11.9 ± 13.61	*p* < 0.05	*p* < 0.05	*p* > 0.05
*p*	*p* < 0.05	*p* < 0.05	*p* < 0.05			
%PPD > 6 mm IL+	5.64 ± 5.16	2.33 ± 3.44	1.37 ± 2.19	*p* < 0.05	*p* < 0.05	*p* < 0.05
%PPD > 6 mm IL−	7.45 ± 13.39	3.24 ± 8.94	2.36 ± 6.97	*p* < 0.05	*p* < 0.05	*p* > 0.05
*p*	*p* > 0.05	*p* > 0.05	*p* > 0.05			

FMPI—full mouth plaque index; FMBOP—full mouth bleeding on probing index; PPD—probing pocket depth; CAL—clinical attachment loss; P—differences within group; *p*—differences between groups.

## Data Availability

The data presented in this study are available on request from the corresponding author.

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
