# Peer review of "Effects of Interleukin-1 Genotype on the Clinical Efficacy of Non-Surgical Periodontal Treatment of Polish Patients with Periodontitis"

_biomedicines, 2023, doi:10.3390/biomedicines11020456_

Round 1
Reviewer 1 Report
Dear Authors,
The reviewer really appreciates the efforts of the authors to conduct this study which has a good clinical significance. However, there are lots of scopes to improve the quality of the manuscript. The reviewer would like to suggest the following revision in the manuscript to make it suitable for publication.
The abstract needs to be re-structured by shortening the methodology and adding highlighted results and a clear conclusion/ outcome of the study.
The Authors have cited 3-6 articles for a single statement. The reviewer’s suggestion would be to use 1 or maximum 2 articles that precisely refer to the statement. Some very references need to be replaced with the recent study stating the fact, for example references 19,34.
I would suggest this paper :
Interferon crevicular fluid profile and correlation with periodontal disease and wound healing: A systemic review of recent data
The methodology needs to be revised in a more organized way by adding a sub-heading for each step for clear understanding.
Although the authors clearly mention the inclusion and exclusion criteria, the method of calculating the sample size is missing in the methodology.
Although the author mention comparison between groups, the statistical analysis (p-value) and correlation between factors (R-value) is missing in the result section.
The conclusion section needs to be revised with a more clear and summarized outcome of the study
I believe that your manuscript would have much more relevance after suggested improvements.
Author Response
Thank you very much for the revision of our manuscript and all comments. The manuscript has been revised according to all your suggestion.
- The abstract has been re-structured
- According to suggestion the part of references has been changed. The number of citations has been improved.
- The methodology has been revised. The method of calculating the sample size was added. To determine the minimal sample size we used G-power 3.1.9.4 software. As the most important analysis was to determinate the impact of treatment process measured by probing depth reduction and restored CAL attachment, the analysis was performed for Wilcoxon rank tests. We assumed moderate size of effect ( r=0,27, equal to d=0,5), standard alpha 0,05, test power 0,85 and the minimal sample size was N=30.
- On the result section, the statistical analysis (p-value and correlation between factors (R-value) was added
- The conclusion section has been revised
I hope the revised version will meet the expectations/
Reviewer 2 Report
The content of the submitted manuscript is good but the presentation way of current form is not fulfilling the journal requirements. Modification is needed to consider for publication.

Author Response
Thank you very much for the revision of our manuscript and all comments. The manuscript has been revised according to all suggestions.
- The title has been modified
- The abstract has been re-structured
- In the introduction, we have supplemented the explanation of the necessity of this study. We described the relationship between periodontal disease and systemic diseases. We also added the null hypothesis .
- To determine the minimal sample size we used the G-Power 3.1.9.4 software. As the most important analysis was to determinate the impact of treatment process measured by probing depth reduction and restored CAL attachment, the analysis was performed for Wilcoxon rank tests. We assumed m moderate size of effect (r=0,27, equal to d=0,5), standard alpha 0,05, test power 0,85 and the minimal sample size was N=30.
- To asses potential biases, all measurements were performed by an experienced and calibrated clinician who was blinded with the respect to the clinical intervention. The test subjects was calibrated at each stage.
I hope this revised version will meet the expectations.
Round 2
Reviewer 1 Report
Dear Authors,
The paper has revised according to suggestions of the reviewers
Reviewer 2 Report
The manuscript has been improved